# Multi-label emotion classification of Urdu tweets

Noman Ashraf[1], Lal Khan[2], Sabur Butt[1], Hsien-Tsung Chang[2,3,4], Grigori Sidorov[1] and Alexander Gelbukh[1]

[1] CIC, Instituto Politécnico Nacional, Mexico City, Mexico
[2] Department of Computer Science and Information Engineering, Chang Gung University, Taoyuan, Taiwan
[3] Artificial Intelligence Research Center, Chang Gung University, Taoyuan, Taiwan
[4] Department of Physical Medicine and Rehabilitation, Chang Gung Memorial Hospital, Taoyuan, Taiwan

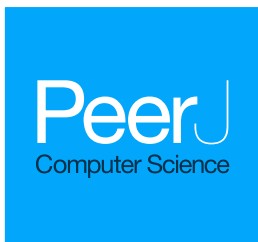

## ABSTRACT

Urdu is a widely used language in South Asia and worldwide. While there are similar datasets available in English, we created the first multi-label emotion dataset consisting of 6,043 tweets and six basic emotions in the Urdu Nastalíq script. A multi-label (ML) classification approach was adopted to detect emotions from Urdu. The morphological and syntactic structure of Urdu makes it a challenging problem for multi-label emotion detection. In this paper, we build a set of baseline classifiers such as machine learning algorithms (Random forest (RF), Decision tree (J48), Sequential minimal optimization (SMO), AdaBoostM1, and Bagging), deep-learning algorithms (Convolutional Neural Networks (1D-CNN), Long short-term memory (LSTM), and LSTM with CNN features) and transformer-based baseline (BERT). We used a combination of text representations: stylometric-based features, pre-trained word embedding, word-based n-grams, and character-based n-grams. The paper highlights the annotation guidelines, dataset characteristics and insights into different methodologies used for Urdu based emotion classification. We present our best results using micro-averaged F1, macro-averaged F1, accuracy, Hamming loss (HL) and exact match (EM) for all tested methods.

## INTRODUCTION

Twitter is a micro blogging platform which is used by millions daily to express themselves, share opinions, and to stay informed. Twitter is an ideal platform for researchers for years to study emotions and predict the outcomes of experimental interventions (*Mohammad & Bravo-Marquez, 2017*; *Mohammad et al., 2015*). Studying emotions in text helps us to understand the behaviour of individuals (*Plutchik, 1980*, *2001*; *Russell & Mehrabian, 1977*; *Ekman, 1992*) and gives us the key to people's feelings and perceptions. Social media text can represent various emotions: happiness, anger, disgust, fear, sadness, and surprise. One can experience multiple emotions (*Strapparava & Mihalcea, 2007*; *Li et al., 2017*) in a small chunk of text while there is a possibility that text could be emotionless or neutral, making it a challenging problem to tackle. It can be easily categorized as a multi-label classification task where a given text can be about any emotion

Corresponding author
Noman Ashraf,
noman@nlp.cic.ipn.mx

simultaneously. Emotion detection in its true essence is a multi-label classification problem since a single sentence may trigger multiple emotions such as anger and sadness. This increases the complexity of the problem and makes it more challenging to classify in a textual setting.

While there are multiple datasets available for multi-label classification in English and other Europian languages, low resource language like Urdu still requires a dataset. The Urdu language is the combination of Sanskrit, Turkish, Persian, Arabic and recently English making it even more complex to identify the true representation of emotions because of the morphological and syntactic structure *Adeeba & Hussain (2011)*. However, the structural similarities of Urdu with Hindi and other South Asian languages make it resourceful for the similar languages. Urdu is the national language of Pakistan that is spoken by more than 170 million people worldwide as first and second language (https://www.ethnologue.com/language/urd). Needless to say, Urdu is also widely used on social media using right to left Nastalíq script.

Therefore, a multi-label emotion dataset for Urdu is long due and needed for understanding public emotions, especially applicable in natural language applications in disaster management, public policy, commerce, and public health. It should also be noted that emotion detection directly aids in solving other text related classification tasks such as sentiment analysis (*Khan et al., 2021*), human aggressiveness and emotion detection (*Bashir et al., 2019*; *Ameer et al., 2021*), humor detection (*Weller & Seppi, 2019*), question answering and fake news detection (*Butt et al., 2021a*; *Ashraf et al., 2021*), depression detection (*Mustafa et al., 2020*), and abusive and threatening language detection (*Ashraf, Zubiaga & Gelbukh, 2021*; *Ashraf et al., 2020*; *Butt et al., 2021b*; *Amjad et al., 2021*).

We created a Nastalíq Urdu script dataset for multi-label emotion classification consisting of 6,043 tweets using Ekman's six basic emotions (*Ekman, 1992*). The dataset is divided into the train and test split which is publicly available along with the evaluation script. The task requires you to classify the tweet as one, or more of the six basic emotions which is the best representation of the emotion of the person tweeting. The paper presents machine-learning and neural baselines for comparison and shows that out of the various machine- and deep-learning algorithms, RF performs the best and gives macro-averaged $F_1$ score of 56.10%, micro-averaged $F_1$ score of 60.20%, and M1 accuracy of 51.20%.

The main contributions of this research are as follows:

- Urdu language dataset for multi-class emotion detection, containing six basic emotions (anger, disgust, joy, fear, surprise, and sadness) (publicly available; see a link below);
- Baseline results of machine-learning algorithms (RF, J48, DT, SMO, AdaBoostM1, and Bagging) and deep-learning algorithms (1D-CNN, LSTM, and LSTM with CNN features) to create a benchmark for multi-label emotion detection using four modes of text representations: word-based *n*-grams, character-based *n*-grams, stylometry-based features, and pre-trained word embeddings.

The rest of the paper is structured as follows.

"Related Work" explains the related work on multi-label emotion classification datasets and techniques. "Multiple-feature Emotion Detection Model" discusses the methodology including creation of the dataset. "Baseline" presents evaluation of our models. "Result Analysis" analyzes the results. "Conclusion and Future Work" concludes the paper and potential highlights for the future work.

# RELATED WORK

Emotion detection has been extended across a number of overlapping fields. As a result, there are a number of publicly available datasets for emotion detection.

## Emotion datasets

EmoBank (*Buechel & Hahn, 2017*) is an English *corpus* of 10,000 sentences using the valence arousal dominance (VAD) representation format annotated with dimensional emotional metadata. EmoBank distinguishes between emotions of readers and writers and is built upon multiple genres and domains. A subset of EmoBank is bi-representationally annotated on Ekman's basic emotions which helps it in mapping between both representative formats. Affective text *corpus* (*Strapparava & Mihalcea, 2007*) is extracted from news websites (Google News, Cable News Network *etc.*) to provide Ekman's emotions (*e.g.*, joy, fear, surprise), valence (positive or negative polarity) and explore the connection between lexical semantics and emotions in news headlines. The emotion annotation is set to [0, 100] where 100 is defined as maximum emotional load and 0 indicates completely missing emotions. Annotations for valence are set to [−100, 100] in which 0 signifies neutral headline, −100 and 100 represent extreme negative and positive headlines, respectively. DailyDialog (*Li et al., 2017*) is a multi-turn dataset for human dialogue. It is manually labelled with emotion information and communication intention and contains 13,118 sentences. The paper follows the six main Ekman's emotions (fear, disgust, anger, and surprise *etc.*) complemented by the "no emotion" category. Electoral Tweets is another dataset (*Mohammad et al., 2015*) which obtains the information through electoral tweets to classify emotions (Plutchik's emotions) and sentiment (positive/ negative). The dataset consists of over 100,000 responses of two questionnaires taken online about style, purpose, and emotions in electoral tweets. The tweets were annotated *via* Crowdsourcing.

The Emotional Intensity (*Mohammad & Bravo-Marquez, 2017*) dataset was created to detect the writers emotional intensity of emotions. The dataset consists of 7,097 tweets where the intensity is analysed by best-worst scaling (BWS) technique. The tweets were annotated with intensities of sadness, fear, anger, and joy using Crowdsourcing. The Emotion Stimulus dataset (*Ghazi, Inkpen & Szpakowicz, 2015*) identifies the textual cause of emotion. It consists of the total number of 2,414 sentences out of which 820 were annotated with both emotions and their cause, while 1,594 were annotated just with emotions. The Grounded Emotions dataset (*Liu, Banea & Mihalcea, 2017*) was designed to study the correlation of users' emotional state and five types of external factors namely user predisposition, weather, social network, news exposure, and timing. The dataset was

**Table 1 Comparison of state-of-the-art in multilabel emotion detection.**

| Link | Size | Language | Data source | Composition |
|---|---|---|---|---|
| EmoBank | 10,000 | English | MASC *Ide et al. (2010)* + SE07 *Strapparava & Mihalcea (2007)* | VAD |
| Affective Text | 1,250 | English | News websites (*i.e.* Google news, CNN) | Ekmans emotions + valence indication (positive/negative). |
| DailyDialog | 13,118 | English | Dialogues from human conversations | Ekman's emotion + No emotion |
| Electoral Tweets | 100,000 | English | Twitter | Plutchik's emotions + sentiment (positive/negative) |
| EmoInt | 7,097 | English | Twitter | Intensities of sadness, fear, anger, and joy |
| Emotion Stimulus | 2,414 | English | FrameNets annotated data | Ekman's emotions and shame |
| Grounded Emotions | 2,557 | English | Twitter | Emotional state (happy or sad) + five types of external factors namely user predisposition, weather, social network, news exposure, and timing |
| Fb-Valence-Arousal | 2,895 | English | Facebook | valence (sentiment) + arousal (intensity) |
| Stance Sentiment Emotion *Corpus* | 4,868 | English | Twitter | Plutchik's emotions |

built upon social media and contains 2,557 labelled instances with 1,369 unique users. Out of these, 1,525 were labelled as happy tweets and 1,032 were labelled as sad tweets. One aspect of sentiment and emotion-related tasks is neutrality in texts. Neutrality often contains ambiguity and a lack of information. Hence, neutrality needs specific characterization to empower models designed for understanding sentiments. A weighted aggregation method for neutrality (*Valdivia et al., 2018*) showed how neutrality is a key in robust sentiment classification. Ambivalence is a phenomenon that includes both negative and positive valenced components towards an action, person or object and hence directly correlates with the sentiment level tasks. An approach for ambivalence handling in texts can be seen in *Wang, Ho & Cambria (2020)*, where the authors used Mixed-Negative, Mixed-Neutral and Mixed-Positive for ambivalence handling. Later, the first step was used for multi-level fine-scaled sentiment analysis.

The Fb-Valence-Arousal dataset (*Preotiuc-Pietro et al., 2016*) consists of 2,895 social media posts collected to train models for valence and arousal. It was annotated by two psychologically trained persons on two separate ordinal nine-point scales with valence (sentiment) or arousal (intensity). The time interval was the same for every message with distinct users. Lastly, the Stance Sentiment Emotion *Corpus* (SSEC) dataset (*Schuff et al., 2017*) is an extension of the SemEval 2016 dataset with a total number of 4,868 tweets. It was extended to enable a relation between annotation layers (sentiment, emotion and stance). Plutchik's fundamental emotions were used for annotation by expert annotators. The distinct feature of this dataset is that they published individual information for all annotators. A comprehensive literature review is summarized in Table 1. Although we have taken an English language emotion data set for comparison, many low resource languages have been catching up in emotion detection tasks in text (*Kumar et al., 2019*;

*Arshad et al., 2019*; *Plaza del Arco et al., 2020*; *Sadeghi, Khotanlou & Rasekh Mahand, 2021*; *Tripto & Ali, 2018*).

XED is a fine-grained multilingual emotion dataset introduced by *Öhman et al. (2020)*. The collection comprises human-annotated Finnish (25 k) and English (30 k) sentences, as well as planned annotations for 30 other languages, bringing new resources to a variety of low-resource languages. The dataset is annotated using Plutchik's fundamental emotions, with neutral added to create a multilabel multiclass dataset. The dataset is thoroughly examined using language-specific BERT models and SVMs to show that XED performs on par with other similar datasets and is thus a good tool for sentiment analysis and emotion recognition.

The examples of annotated dataset for emotion classification show that the difference lies between annotation schemata (*i.e.*, VAD or multi-label discreet emotion set), the domain of the dataset (*i.e.*, social news, questionnaire, and blogs *etc.*), the file format, and the language. Some of the most popular datasets released in the last decade to compare and analyze in Table 1. For a more comprehensive review of existing datasets for emotion detection, we refer the reader to *Murthy & Kumar (2021)*.

## Approaches to emotion detection

Sentiment classification has been around for decades and has been the centre of the research in natural language processing (NLP) (*Zhang, Wang & Liu, 2018*). Emotion detection and classification became naturally the next step after sentiment task, while psychology is still determining efficient emotion models (*Barrett et al., 2018*; *Cowen & Keltner, 2018*). NLP researchers embraced the most popular (*Ekman, 1992*; *Plutchik, 1980*) definitions and started working on establishing robust techniques. In the early stages, emotion detection followed the direction of Ekman's model (*Ekman, 1992*) which classifies emotions in six categories (disgust, anger, joy, fear, surprise, and sadness). Many of the recent work published in emotion classification follows the wheel of emotions (*Plutchik, 1980*, *2001*) which classifies emotions as (fear-anger, disgust-trust, joy-sadness, and surprise-anticipation) and *Plutchik (1980)* eight basic emotions (Ekman's emotion plus anticipation and trust) or the dimensional models making a vector space of linear combination affective states (*Russell & Mehrabian, 1977*).

Emotion text classification task has been divided into two methods: rule-based and machine-learning based. Famous examples stemming from expert notation can be SentiWordNet (*Esuli & Sebastiani, 2007*) and WordNet-Affect (*Strapparava & Valitutti, 2004*). Linguistic inquiry and word count (LIWC) (*Pennebaker, Francis & Booth, 2001*) is another example assigning lexical meaning to psychological tasks using a set of 73 lexicons. NRC word-emotion association lexicon (*Mohammad, Kiritchenko & Zhu, 2013*) is also an available extension of the previous works built using eight basic emotions (*Plutchik, 1980*), whereas the values of VAD (*Russell & Mehrabian, 1977*) were also used for annotation (*Warriner, Kuperman & Brysbaert, 2013*). Rule-based work was superseded by supervised feature-based learning using variations of features such as word embeddings, character *n*-grams, emoticons, hashtags, affect lexicons, negation and punctuation (*Jurgens et al., 2012*; *Aman & Szpakowicz, 2007*; *Alm, Roth & Sproat, 2005*).

As part of emotional computing, emotion detection is commonly employed in the educational domain. *Halim, Waqar & Tahir (2020)* presented a methology for detecting emotion in email messages. The framework is built on autonomous learning techniques and uses three machine learning classifiers such as ANN, SVM and RF and three feature selection algorithms to identify six (neutral, happy, sad, angry, positively surprised, and negatively surprised) emotional states in the email text. Study (*Plaza-del Arco et al., 2020*) offered research of multiple machine learning algorithms for identifying emotions in a social media text. The findings of experiments with knowledge integration of lexical emotional resources demonstrated that using lexical effective resources for emotion recognition in languages other than English is a potential way to improve basic machine learning systems. IDS-ECM, a model for predicting emotions in textual dialogue, was also presented in *Li, Li & Wang (2020)*. Textual dialogue emotion analysis and generic textual emotion analysis were contrasted by the authors. They also listed context-dependence, contagion, and persistence as hallmarks of textual dialogue emotion analysis.

Neural network-based models (*Barnes, Klinger & Schulte im Walde, 2017*; *Schuff et al., 2017*) techniques like bi-LSTM, CNN, and LSTM achieve better results compared to feature-based supervised model *i.e.*, SVM and MaxEnt. The leading method at this point is claimed using bi-LSTM architecture aided by multi-layer self attention mechanism (*Baziotis et al., 2018*). The state-of-the-art accuracy of 59.50% was achieved. In *Hassan, Shaar & Darwish (2021)* examine three approaches: (i) employing intrinsically multilingual models; (ii) translating training data into the target language, and (iii) using a parallel *corpus* that is automatically labelled. English is used as the source language in their research, with Arabic and Spanish as the target languages. The efficiency of various classification models was investigated, such as BERT and SVMs, that have been trained using various features. For Arabic and Spanish, BERT-based monolingual models trained on target language data outperform state-of-the-art (SOTA) by 4% and 5% absolute Jaccard score, respectively. For Arabic and Spanish, BERT models achieve accuracies of 90% and 80% respectively.

One of the exciting studies (*Basiri et al., 2021*) proposed a CNN-RNN Deep Bidirectional Model based on Attention (ABCDM). ABCDM evaluates temporal information flow in both directions utilizing two independent bidirectional LSTM and GRU layers to extract both past and future contexts. Attention mechanisms were also applied to the outputs of ABCDM's bidirectional layers to place more or less focus on certain words. To minimize feature dimensionality and extract position-invariant local features, ABCDM uses convolution and pooling methods. The capacity of ABCDM to detect sentiment polarity, which is the most common and significant task in sentiment analysis, is a key metric of its effectiveness. ABCDM achieves state-of-the-art performance on both long review and short tweet polarity classification when compared to six previously suggested DNNs for sentiment analysis. We also saw attention based methods (*Gan, Wang & Zhang, 2020*; *Basiri et al., 2021*) for sentiment related tasks. An effective deep learning method can be seen *Basiri et al. (2021)* which uses Attention-based Bidirectional CNN-RNN addressing the problems of high feature dimensionality and feature weighting. The model uses bi-directional contexts, position-invariant local features

and pooling mechanisms for sentiment polarity detection to achieve the state of the art results. Another popular approach (*Majumder et al., 2020*) uses conditional random field (CRF) and bidirectional gated recurrent unit (BiGRU) based sequence tagging method for aspect extraction. The approach later concatenates GloVe embeddings with the aspect extracted data as input to the aspect-level sentiment analysis (ALSA) models.

### Research gap

Some of the important work in Roman Urdu sentiment detection is done by multiple researchers (*Mehmood et al., 2019*; *Arshad et al., 2019*); however, to the best of our knowledge, no prior work on multi-label emotion classification exists for the Nastalíq Urdu language. From Table 1, one can observe that no annotated dataset was available for multi-label emotion classification task in Nastalíq script. Detecting Nastalíq script on Twitter requires attention and can further aid in solving problems like abusive language detection, humor detection and depression detection in text. Our motivation was to provide an in-depth feature engineering for the task, describing not only lexical features but also embedding, comparing the performance of these features for Nastalíq script in Urdu. We also saw a lack of comparison between classifiers. Most of the studies used either only machine learning or only deep learning (DL) techniques, while no comparison was done between ML and DL models, whereas, we gave the baseline results for both ML and DL classifiers.

## MULTIPLE-FEATURE EMOTION DETECTION MODEL

The emotion detection model is illustrated in Fig. 1. The figure explains the basic architecture followed for both machine learning and deep learning classifiers. Our model has three main phases: data collection, feature extraction (*i.e.*, character $n$-grams, word $n$-grams, stylometry-based features, and pre-trained word embedding), and emotion detection classification.

"Dataset" explains all the details related to dataset: data crawling, data annotation, and characteristics and standardization while "Feature Representations" talks about features types and features extraction methods. Classification algorithms and methodology thoroughly explained in "Setup and Classifiers".

### Dataset

The multi-label emotion dataset in Urdu is neither available nor has any experiments conducted in any domain. Tweets elucidate the emotions of people as they describe their activities, opinions, and events with the world and therefore is the most appropriate medium for the task of emotion classification. The goal of this dataset is to develop a large benchmark in Urdu for the multi-label emotion classification task. This section describes the challenges confronted during accumulation of a large benchmark Twitter-based multi-label emotion dataset and discusses the data crawling method, data collection requirements, data annotation process and guidelines, inter-annotator agreement, and dataset characteristics and standardization. Figure 2 contains the examples of the dataset.

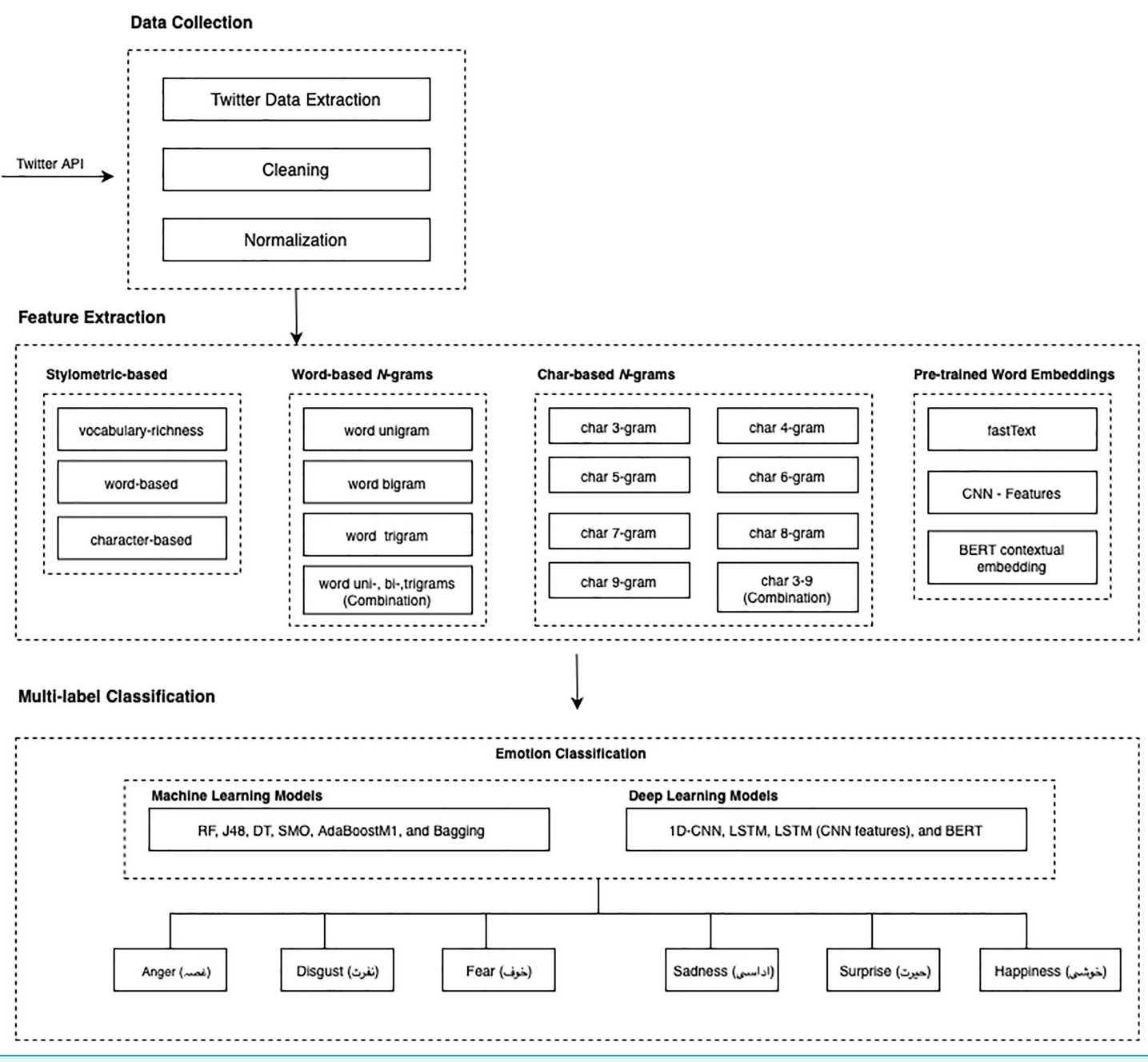

**Figure 1  Multilabel emotion detection model for Urdu language.**

### Data crawling

The dataset was obtained through Twitter and we use Ekman's emotion keywords for the collection of tweets. Twitter developer application programming interface (API) (*Twitter, 2006*) was used and the resulting tweets were collected in a CSV file. The script for the purpose of scrapping was developed in python which was filtered using hashtags, query strings, and user profile name through Twitter rest API.

| Tweets | Anger (غصہ) | Disgust (نفرت) | Fear (خوف) | Sadness (اداسی) | Surprise (حیرت) | Happiness (خوشی) |
|---|---|---|---|---|---|---|
| اب تو لعنت کا ہی مقام رہ گیا ہے اس شہباز گل اور اس کے لیڈرز کے لیے حیرت ہوتی ہے کہ اسط کی ٹوئیٹ<br><br>Now it is a place of curse. It is a surprise for Shahbaz Gul and his leaders to tweet like this | 1 | 1 | 0 | 0 | 1 | 0 |
| دماغ پک گیا ہے نئی تحقیق آئی ہے کہ کرونا سے صحت یاب شخص پھر سے انفیکٹ نہیں ہوتا ربہ روز اچانک سے ڈ<br><br>The brain has matured. New research has shown that a person who recovers from corona is not infected again. | 1 | 1 | 0 | 0 | 1 | 0 |
| مجھے منافق لوگوں سے نفرت<br><br>I hate hypocrites | 0 | 1 | 0 | 0 | 0 | 0 |
| اللہ دور کرے اداسی<br><br>May Allah remove the sadness | 0 | 0 | 0 | 1 | 0 | 0 |
| سالگرہ مبارک ہو حسین مجھے خوشی ہے کہ مجھے آپ کی زندگی میں ایک اور سال گزارنا پڑا اور بہت سال مجھے پیار ہے آپ اپن<br><br>Happy birthday hussain i am glad i had to spend another year in your life and many years i love you your | 0 | 0 | | 0 | 0 | 1 |
| وزیر اعظم صاحب خدا کا خوف کرو ڈاکٹرز اپیل کر رہے ہں کہ پاکستان میں کرونا کی تباہی پھیل گ ہے عوام اور ڈاکٹر اس وبا<br><br>Prime Minister, fear God, doctors are appealing that the destruction of Karuna will spread in Pakistan. | 0 | 0 | 1 | 1 | 1 | 0 |

**Figure 2 Examples in our dataset (translated by Google).**

**Table 2 Distribution of emotions in the dataset.**

| Emotions | Train | Test |
|---|---|---|
| Anger (غصہ) | 833 | 191 |
| Disgust (نفرت) | 756 | 203 |
| Fear (خوف) | 594 | 184 |
| Sadness (اداسی) | 2,206 | 560 |
| Surprise (حیرت) | 1,572 | 382 |
| Happiness (خوشی) | 1,040 | 278 |

For each emotion, the maximum of two thousand tweets were extracted which were later refined and shrunk per keyword based on tweet quality and structure. All the tweets with multiple languages (*i.e.*, Arabic and Persian) were eliminated from the dataset and only the purest Urdu tweets were kept. The total collected tweets, in the end, were twelve thousand. Table 2 mentions the final distribution of tweets per label. The features mentioned in each example of tweet included tweetid, tweet, hashtags, username, date, and time. The dataset is publicly available on GitHub (https://github.com/Noman712/Mutilabel_Emotion_Detection_Urdu/tree/master/dataset).

## Data annotation

As mentioned previously, the Twitter hashtags were used for extracting relevant tweets of a particular emotion. However, since a tweet can contain multiple emotions, the keywords

alone cannot be a reliable method for annotation. Therefore, data annotation standards were prepared for expert annotators to follow and maintain consistency throughout the task.

- Anger (غصہ) also includes annoyance and rage can be categorized as a response to a deliberate attempt of anticipated danger, hurt or incitement.
- Disgust (نفرت) also including anxiety, panic and horror is an emotion in a text which can be seen triggered through a potential cumbersome situation or danger.
- Sadness (اداسی) also including pensiveness and grief is triggered through hardship, anguish, feeling of loss, and helplessness.
- Surprise (حیرت) also including distraction and amazement is an emotion which is prompted by an unexpected occurrence.
- Happiness (خوشی) also including contentment, pride, gratitude and joy is an emotion which is seen as a response to well-being, sense of achievement, satisfaction, and pleasure.

### Annotation guidelines
The following guidelines were set for the annotation process of the dataset:

- Three specialised annotators in the field of Urdu were selected. Both annotators had the minimum qualification of Masters in Urdu language making them the most suitable persons for the job.
- Complete dataset was provided to two of the annotators and they were asked to classify the tweets in one or multiple emotion labels with a minimum of one and maximum of six emotions. The existing emotions were labelled as 1 under each category and the rest were marked 0.
- The annotator's results were observed and analysed after every 500 tweets to ensure the credibility and correct pattern of annotation.
- The annotators were asked to identify emojis in a tweet with their corresponding labels. They were informed of the possibility of varying context between emojis and text. In such a case, multiple suited labels were selected to portray multiple or mix emotions.
- Major conflicts where at least one category was labelled differently by the previous two annotators were identified and the labelled dataset for the conflicting tweets was resolved by the third annotator.

Inter-annotator agreement (IAA) was computed using Cohan's Kappa Coefficient (*Cohen, 1960*). We achieved kappa coefficient of 71% which shows the strength of our dataset.

### Dataset characteristics and standardization
UrduHack (https://pypi.org/project/urduhack/) was used to normalize the tweets. Urdu text has diacritics (a glyph added to an alphabet for pronunciation) which needs to be removed. For both word and character level normalization, we removed the diacritics,

**Table 3 Statistics based on the train and test dataset.**

| Dataset | Tweets | Words | Avg. Word | Char | Avg. Char | Vocab |
|---|---|---|---|---|---|---|
| All | 6,043 | 44,525 | 9.24 | 224,806 | 46.65 | 14,101 |
| Train | 4,818 | 44,525 | 9.24 | 224,806 | 46.65 | 9,840 |
| Test | 1,225 | 11,425 | 9.32 | 57,658 | 47.06 | 4,261 |

added spaces after digits, punctuation marks, and stop words (https://github.com/urduhack/urdu-stopwords/blob/master/stop_words.txt) form the data. For character level normalization, Unicode were assigned to each character. Table 2 shows the frequently occurring emotions. In a multi-label setting, several emotions appear in a tweet, hence, the number of emotions exceed the number of tweets. The emotion anger (غصہ) is seen to be the most common emotion used in the tweets. Meanwhile Table 3 shows the statistics of the tweets after normalization in train and test dataset. The entire dataset has the vocabulary of 14,101 words while each tweet average length is 9.24 words and 46.65 characters.

# BASELINE

## Feature representations

Four types of text representation were used: character *n*-grams, word *n*-grams, stylometric features, and pre-trained word embeddings.

### Count based features

Character *n*-grams and token *n*-grams were used as count-based features. We generated word uni-, bi-, and trigrams and character *n*-grams from trigrams to ninegrams. Term frequency-inverse document frequency (TF-IDF), a feature weighting technique on count-based features[1] was also used. Scikit-Learn (https://scikit-learn.org/stable/) was used for the extraction of all features.

### Stylometry based features

The second set was stylometric based features (*Lex, Juffinger & Granitzer, 2010*; *Grieve, 2007*) which included 47 character-based features, 11 word-based features and 6 vocabulary-richness based features. Stylometry based features are used to analyze literary style in emotions (*Anchiêta et al., 2015*), whereas, vocabulary richness based features are used to capture individual specific vocabulary (*Milička & Kubát, 2013*).

The character-based features are as follows:

- Number of apostrophe, ampersands, asterisks, at the rate signs, brackets, characters without spaces, colons, commas, counts, dashes, digits, dollar signs, ellipsis, equal signs, exclamation marks, greater and less than signs, left and right curly braces, left and right parenthesis, left and right square brackets, full stops, multiple question marks, percentage signs, plus signs, question marks, tilde, underscores, tabs, slashes, semicolons, single quotes, vertical lines, and white spaces;
- Percentage of commas, punctuation characters, and semi-colons;

[1] We use the following parameters: use_idf = True, smooth_idf = True, and number of features 1,000.

- Ratio by *N* (where *N* = total no of characters in Urdu tweets) of white spaces by *N*, digits by *N*, letters by *N*, special characters by *N*, tabs by *N*, upper case letter and characters by *N*.

The word-based features are as follows:

- Average of word length, sentence length, words per paragraph, sentence length in characters, and number of sentences,
- Number of paragraphs,
- Ratio of words with length 3 and 4,
- Percentage of question sentences,
- Total count of unique words and the total number of words.

The vocabulary-richness based features are as follows:

- BrunetWMeasure,
- HapaxLegomena,
- HonoreRMeasure,
- SichelSMeasure,
- SimpsonDMeasure,
- uleKMeasure.

### Pre-trained word embeddings

Word embeddings were extracted from the tweets using fastText (https://fasttext.cc/) with 300 vector space dimensions per word. Only fastText was used as it contains the most dense vocabulary for Urdu Nastalíq script. Since the text was informal social media tweets, it was highly probable that some words are missing in the dictionary. In that condition, we randomly assigned all 300 dimensions with a uniform distribution in [−0.1, 0.1].

### Setup and classifiers

We treated multi-label emotion detection problem as a supervised classification task. Our goal was to predict multiple emotions from the six basic emotions. We used tenfold cross validation for this task which ensures the robustness of our evaluation. The tenfold cross validation takes 10 equal size partitions. Out of 10, one subset of the data is retained for testing and the rest for training. This method is repeated 10 times with each subset used exactly once as a testing set. The 10 results obtained are then averaged to produce estimation. For our emotion detection problem binary relevance and label combination (LC) transformation methods were used along with various machine- and deep-learning algorithms: RF, J48, DT, SMO, AdaBoostM1, Bagging, 1D CNN, and LSTM. As evidently these algorithms perform extremely well for several NLP tasks such as sentiment analysis, and recommendation systems (*Kim, 2014*; *Hochreiter & Schmidhuber, 1997*; *Breiman, 2001*; *Kohavi, 1995*; *Sagar, Jhaveri & Borrego, 2020*; *Panigrahi et al., 2021a*, *2021b*).

We used several machine learning algorithms to test the performance of the dataset namely: RF, J48, DT, SMO, AdaBoostM1 and Bagging. AdaBoostM1 (*Freund & Schapire, 1996*) is a very famous ensemble method which diminishes the hamming loss by creating models repetitively and assigning more weight to misclassified pairs until the maximum model number is not achieved. RF is another ensemble classification method based on trees which is differentiated by bagging and distinct features during learning. It is robust as it overcomes the deficiencies of decision trees by combining the set of trees and input variable set randomization (*Breiman, 2001*). Bagging (Bootstrap Aggregation) (*Breiman, 1996*) is implemented which aggregates multiple machine learning predictions and reduces variance to give a more accurate result. Lastly, SMO (*Hastie & Tibshirani, 1998*) which decomposes multiple variables into a series of sub-problems and optimizes them as mentioned in the previous studies. DT and J48 were also tested as described in the papers (*Salzberg, 1994*; *Kohavi, 1995*); however, they were unable to achieve substantial results. For machine learning algorithms we used MEKA (http://meka.sourceforge.net) default parameters to provide the baseline scores.

We experimented with our multi-label classification task with two deep learning models: 1-dimensional convolutional neural network (1D CNN) and long short-term memory (LSTM). We used LSTM (*Hochreiter & Schmidhuber, 1997*) which is the enhanced version of the recurrent neural network with the difference in operational cells and enables it to keep or forget information increasing the learning ability for long-time sequence data. CNN (*Kim, 2014*) takes the embeddings vector matrix of tweets as input with the multi-label distribution and then passes through filters and hidden layers. We used Adam optimizer, categorical cross-entropy as a loss function, softmax activation function on the last layer, and dropout layers of 0.2 in both LSTM and 1D-CNN. Figure 3 shows the architecture of 1D-CNN while Fig. 4 shows the architecture of LSTM model. Table 4 shows the fully connected layers and their parameters for 1D-CNN and LSTM.

The tweets were passed as word piece embeddings which were later channelled into a sequence. Keras (https://keras.io/) and Pytorch (https://pytorch.org) framework were used for the implementation of all these algorithms. For additional details on the experiments, please review the publicly available code (https://github.com/Noman712/Mutilabel_Emotion_Detection_Urdu/tree/master/code).

BERT (*Devlin et al., 2018*) has proven in multiple studies to have a better sense of flow and language context as it is trained bidirectionally with an attention mechanism. We used the following BERT parameters: max-seq-length = 64, batch size = 32, learning rate = 2e−5, and num-train-epochs = 2.0. We used 0.1 dropout probability, 24 hidden layers, 340 M parameters and 16 attention heads respectively.

## Metrics and evaluation

To evaluate multi-label emotion detection, we used multi-label accuracy, micro-averaged $F_1$ and macro-averaged $F_1$. Multi-label accuracy in the emotion classification considers the subsets of the actual classes for prediction as a mis-classification is not hard wrong or right *i.e.*, predicting two emotions correctly rather than declaring no emotion. For multi-
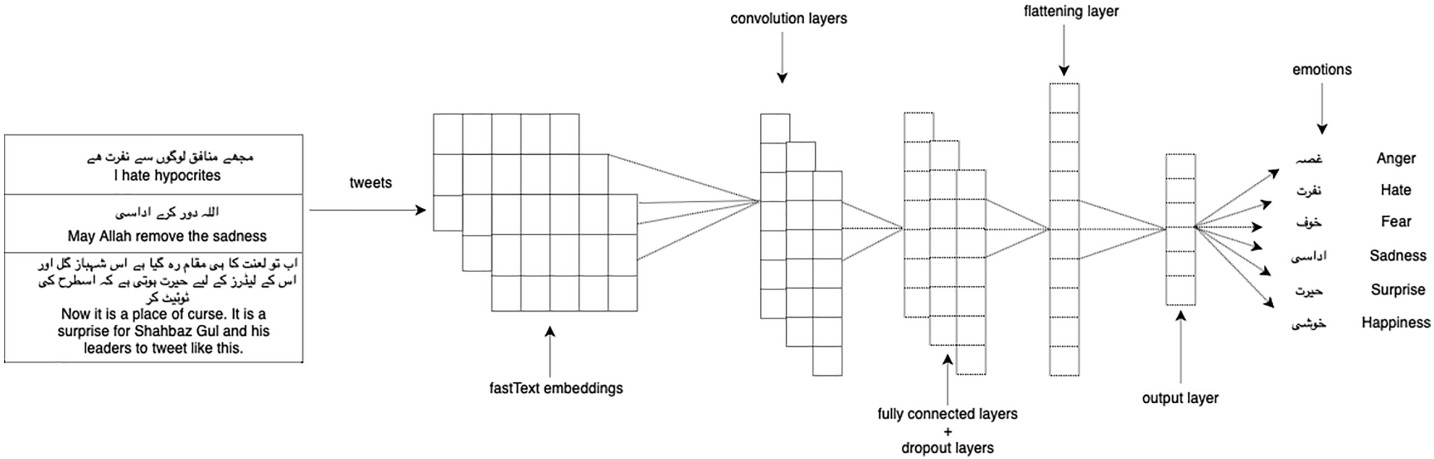

**Figure 3 1D-CNN model architecture.**

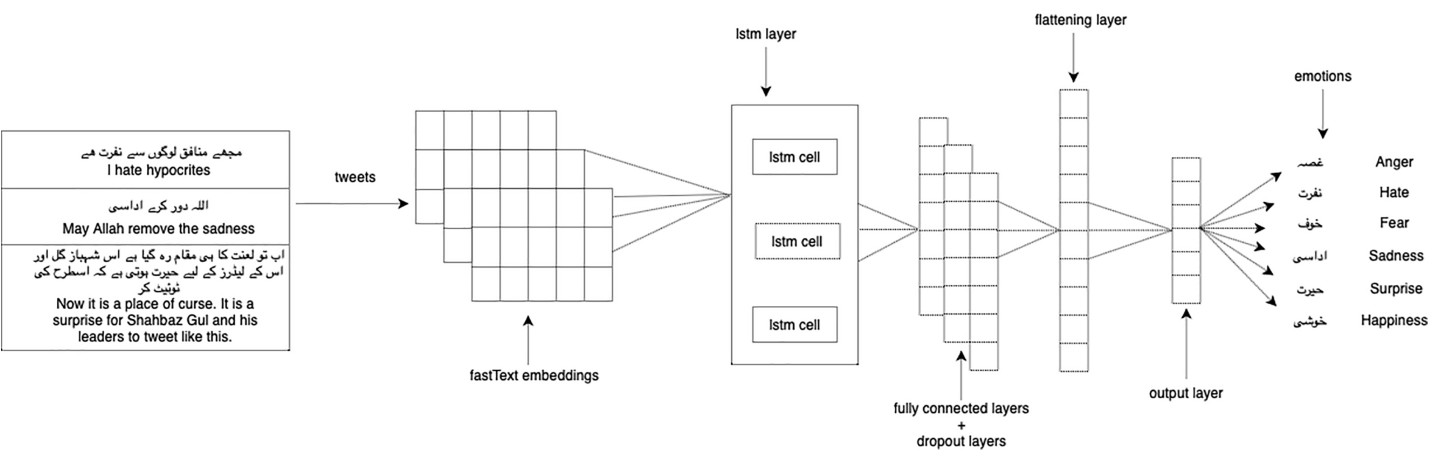

**Figure 4 LSTM model architecture.**

label accuracy, we considered one or more gold label measures compared with obtained emotion labels or set of labels against each given tweet. We took the size of the intersection of the predicted and gold label sets divided by the size of their union and then averaged it over all tweets in the dataset.

Micro-averaging, in this case, will take all True Positives (TP), True Negatives (TN), False Positives (FP), and False Negatives (FN) individually for each tweets label to calculate precision and recall. The mathematical equations of micro-averaged $F_1$ are provided in 1,2,3 respectively:

$$P_{micro} = \frac{\sum_{e \in E} number\ of\ c(e)}{\sum_{e \in E} number\ of\ p(e)},$$

**Table 4 Deep learning parameters for 1D-CNN and LSTM.**

| Parameter | 1D-CNN | LSTM |
|---|---|---|
| Epochs | 100 | 150 |
| Optimizer | Adam | Adam |
| Loss | categorical crossentropy | categorical crossentropy |
| Learning Rate | 0.001 | 0.0001 |
| Regularization | 0.01 | – |
| Bias Regularization | 0.01 | – |
| Validation Split | 0.1 | 0.1 |
| Hidden Layer 1 Dimension | 16 | 16 |
| Hidden Layer 1 Activation | tanh | tanh |
| Hidden Layer 1 Dropout | 0.2 | – |
| Hidden Layer 2 Dimension | 32 | 32 |
| Hidden Layer 2 Activation | tanh | tanh |
| Hidden Layer 2 Dropout | 0.2 | – |
| Hidden Layer 3 Dimension | – | 64 |
| Hidden Layer 3 Activation | – | tanh |
| Hidden Layer 3 Dropout | – | – |

$$R_{micro} = \frac{\sum_{e \in E} number\ of\ c(e)}{\sum_{e \in E} number\ of\ (e)},$$

$$F1_{micro} = \frac{2 \times P_{micro} \times R_{micro}}{P_{micro} + R_{micro}}.$$

The $c(e)$ notation denotes the number of samples correctly assigned to the label $e$ out of sample $E$, $p(e)$ defines the number of samples assigned to $e$, and $(e)$ represents the number of actual samples in $e$. Thus, P-micro is the micro-averaged precision score, and R-micro is the micro-averaged recall score. Macro-averaging, on the other hand, uses precision and recall based on different emotion sets, calculating the metric independently for each class treating all classes equally. Then, $F_1$ was calculated as mentioned in the equation for both. The mathematical equations of macro-averaged $F_1$ are provided in 1,2,3 respectively:

$$P_e = \frac{\sum_{e \in E} number\ of\ c(e)}{\sum_{e \in E} number\ of\ p(e)},$$

$$R_e = \frac{\sum_{e \in E} number\ of\ c(e)}{\sum_{e \in E} number\ of\ (e)},$$

$$F_e = \frac{2 \times P_e \times R_e}{P_e + R_e},$$

$$F1_{macro} = \frac{1}{|E|} \sum_{e \in E} F_e.$$

**Table 5 Best results for multi-label emotion detection using word _n_-gram features.**

| Features | MLC | SLC | Acc. | EM | HL | Micro-$F_1$ | Macro-$F_1$ |
|---|---|---|---|---|---|---|---|
| Word N–gram | | | | | | | |
| Word 1-gram | BR | RF | 51.20 | 32.30 | 19.40 | 60.20 | 56.10 |
| Word 2-gram | LC | SMO | 43.60 | 30.30 | 21.70 | 50.20 | 47.50 |
| Word 3-gram | BR | RF | 39.90 | 16.60 | 28.40 | 50.00 | 48.10 |
| Combination of Word N–gram | | | | | | | |
| Word 1–3-gram | BR | AdaBoostM1 | 35.10 | 14.90 | 30.10 | 44.50 | 42.60 |

The exact match equation is mentioned below which explains the percentage of instance whose predicted labels ($P_t$) are exactly matching same the true set of labels ($G_t$).

$$ExactMatch = \frac{1}{|T|}\sum_{i=1}^{T} G_t = P_t$$

The Hamming loss equation mentioned below computes the average of incorrect labels of an instance. Lower the value, higher the performance of the classifier as this is a loss function.

$$HammingLoss = \frac{1}{|TS|}\sum_{i=1}^{T}\sum_{j=1}^{S} G_j^i = P_j^i$$

## RESULT ANALYSIS

We conducted several experiments with detailed insight into our dataset. Table 5 shows the result of each of the baseline machine and deep-learning classifiers using word _n_-grams to detect multi-label emotions from our dataset. Uni-gram shows the best result on RF in combination with a BR transformation method and it achieves 56.10% of macro $F_1$. It outperforms bigram and trigram features. When word uni-, bi-, and trigrams, features are combined, AdaboostM1 gives the best results and obtains 42.60% of macro $F_1$. However, results achieved with combined features are still inferior as compared to individual _n_-gram features. A series of experiments on character _n_-grams were conducted. Results of char 3-gram to char 9-gram are mentioned in Table 6. It shows that RF consistently provides the best results paired with BR on character 3-gram and obtains the macro $F_1$ of 52.70%. It is observed that macro $F_1$ decreases while increasing the number of characters in our features. A combination of character _n_-gram (3–9) achieved the best results using RF with LC, but still lagged behind all individual _n_-gram measures. Overall, word based _n_-gram feature results are very close to each other and achieves better results than most of the char based _n_-gram features.

Table 7 illustrates the results of stylometry-based features which were tested on a different set of feature groups such as character-base, word-base, vocabulary richness and combination of first three features. Word-based feature group depicts the macro $F_1$ of 42.60% which is trained on Adaboost M1 and binary relevance. Lastly, experiments on

**Table 6 Best results for multi-label emotion detection using char *n*-gram features.**

| Features | MLC | SLC | Acc. | EM | HL | Micro-$F_1$ | Macro-$F_1$ |
|---|---|---|---|---|---|---|---|
| Character N-gram | | | | | | | |
| Char 3-gram | BR | RF | 47.20 | 28.20 | 21.10 | 56.60 | 52.70 |
| Char 4-gram | BR | Bagging | 38.60 | 21.70 | 25.60 | 47.30 | 44.60 |
| Char 5-gram | BR | Bagging | 38.30 | 16.50 | 28.80 | 47.90 | 46.30 |
| Char 6-gram | BR | Bagging | 37.80 | 16.90 | 29.30 | 46.30 | 45.50 |
| Char 7-gram | BR | RF | 36.10 | 15.50 | 31.00 | 44.70 | 43.80 |
| Char 8-gram | BR | RF | 34.80 | 11.80 | 31.50 | 45.30 | 43.50 |
| Char 9-gram | BR | RF | 34.80 | 11.80 | 31.50 | 45.10 | 43.40 |
| Combination of Character N-gram | | | | | | | |
| Char 3–9 | LC | RF | 33.60 | 32.90 | 12.10 | 32.30 | 33.90 |

**Table 7 Best results for multi-label emotion detection using stylometry-based features.**

| Features | MLC | SLC | Acc. | EM | HL | Micro-$F_1$ | Macro-$F_1$ |
|---|---|---|---|---|---|---|---|
| Character-based | BR | DT | 33.70 | 10.7 | 31.90 | 44.40 | 42.40 |
| Word-based | BR | AdaBoostM1 | 35.10 | 14.90 | 30.10 | 44.50 | 42.60 |
| Vocabulary richness | BR | AdaBoostM1 | 34.10 | 11.80 | 31.10 | 44.50 | 42.50 |
| All features | BR | AdaBoostM1 | 35.00 | 14.90 | 30.00 | 44.50 | 42.50 |

**Table 8 Best results for multi-label emotion detection using pre-trained word embedding features.**

| Model | Features (dim) | Acc. | EM | HL | Micro-$F_1$ | Macro-$F_1$ |
|---|---|---|---|---|---|---|
| 1D CNN | fastText (300) | 45.00 | 42.00 | 36.00 | 35.00 | 54.00 |
| LSTM | fastText (300) | 44.00 | 42.00 | 35.00 | 32.00 | 55.00 |

deep-learning algorithms such as 1D-CNN, LSTM, LSTM with CNN features show promising results for multi-label emotion detection. LSTM achieves the highest macro $F_1$ score of 55.00% while 1D-CNN and LSTM with CNN features achieve slightly lower macro $F_1$ scores. Tables 8 and 9 show the results of deep-learning algorithms.

Considering four text representations, the best-performing algorithm is RF with BR that trained on uni-gram features achieve macro $F_1$ score of 56.10%. Deep learning algorithms performed well using fastText pre-trained word embeddings and results are consistent in all the experiments.

Notably, machine-learning baseline using word based *n*-gram features achieved highest macro $F_1$ score of 56.10% comparatively to deep-learning baseline that achieved slightly lower $F_1$ score of 55.00% using pre-trained word embeddings. Pre-trained word embedding was not able to obtain the highest results, it might be because fastText does not have all of the vocab for Urdu language and some of the words could be missed as out-of-vocabulary. Therefore, further research is needed for pre-trained word embeddings

**Table 9 Best results for multi-label emotion detection using contextual pre-trained word embedding features.**

| Model | Features (dim) | Acc. | EM | HL | Micro-F$_1$ | Macro-F$_1$ |
|---|---|---|---|---|---|---|
| LSTM | fastText (300), 1D CNN (16) | 46.00 | 35.00 | 36.00 | 34.00 | 53.00 |
| BERT | BERT Contextual Embeddings (768) | 15.00 | 44.00 | 57.00 | 54.00 | 37.00 |

**Table 10 Comparison of state-of-the-art results in multi-label emotion detection.**

| Reference | Model | Features | Accuracy | Micro-F$_1$ | Macro-F$_1$ | HL |
|---|---|---|---|---|---|---|
| *Ameer et al. (2021)* | RF | n-gram | 45.20 | 57.30 | 55.90 | 17.90 |
| *Zhang et al. (2020)* | MMS2S | – | 47.50 | – | 56.00 | 18.30 |
| *Samy, El-Beltagy & Hassanien (2018)* | C-GRU | AraVec, word2vec | 53.20 | 49.50 | 64.80 | – |
| *Ju et al. (2020)* | MESGN | – | 49.4 | – | 56.10 | 18.00 |
| Proposed | 1D CNN | fastText | 45.00 | 35.00 | 54.00 | 36.00 |
| Proposed | RF | word unigram | 51.20 | 60.20 | 56.10 | 19.40 |

and deep-learning approaches that might help to improve the results. Table 10 shows the state of the art results for multi-label emotion detection in English and proves that our baseline results are in line with state-of-the-art work in the machine and deep learning.

## Discussion

In terms of reproducibility, our machine learning algorithm results are much easier to reproduce with MEKA software. It is because default parameters were used to analyze the baseline results. The main challenge for this task is to generate $n$-gram features in a specific .arff format which is the main requirement of this software to run the experiments. For this purpose, we use sklearn library to extract features from the Urdu tweets and then use Python code to convert them into the .arff supported format. The code is publicly available. Hence, academics and industrial environments can repeat experiments by just following the guidelines of the software.

In addition, computational complexity can make the reproducibility challenging of the proposed methods. Few years ago, it was difficult to produce the results as they can take days or weeks, although researchers have access to GPU computing. Classifiers such as Random Forest and Adaboost that are used in this paper can lead to scalability issues. However, scalability can be addressed with appropriate feature engineering and pre-processing techniques in both academia and industry (*Jannach & Ludewig, 2017*; *Linden, Smith & York, 2003*).

## CONCLUSION AND FUTURE WORK

In this research, we created a multi-label emotion dataset in Urdu based on social media which is the first for Urdu Nastalíq script. Data characteristics for Urdu needed in order to refine social media data were defined. Impact of results were shown by conducting experiments, analysing results on stylometric-based features, pre-trained word embedding, word $n$-grams, and character $n$-grams for multi-label emotion detection. Our

experiments concluded that RF combined with BR performed the best with uni-gram features achieving 56.10 micro-averaged $F_1$, 60.20 macro-averaged $F_1$, and 51.20 M1 accuracy. The superiority of machine-learning techniques over neural baselines identified a vacuum for the neural net techniques to experiment. There are several limitations of this work: (1) Reproducibility is one of the major concern because of the computational complexity and scalability of the algorithms such as RF and Adaboost. (2) Another limitation is fastText pre-trained word embeddings does not have all of the vocab for Urdu language, therefore, some of the words could be missed as out-of-vocabulary. As a result, performance of the deep learning classifiers are poor as compared to the machine learning classifiers. Our dataset is expected to meet the challenges of identifying emotions for a wide range of NLP applications: disaster management, public policy, commerce, and public health. In future, we expect to outperform our current results using novel methods, extend emotions, and detect the intensity of emotions in Urdu Nastalíq script.

### Funding
The work was done with support from the Mexican Government through the grant A1-S-47854 of the CONACYT, Mexico and grants 20211784, 20211884, and 20211178 of the Secretaría de Investigación y Posgrado of the Instituto Politécnico Nacional, Mexico. The funders had no role in study design, data collection and analysis, decision to publish, or preparation of the manuscript.

### Grant Disclosures
The following grant information was disclosed by the authors:
CONACYT: A1-S-47854.
Secretaría de Investigación y Posgrado of the Instituto Politécnico Nacional, Mexico: 20211784, 20211884 and 20211178.

### Competing Interests
The authors declare that they have no competing interests.

### Author Contributions
- Noman Ashraf conceived and designed the experiments, performed the experiments, analyzed the data, performed the computation work, prepared figures and/or tables, and approved the final draft.
- Lal Khan conceived and designed the experiments, performed the experiments, analyzed the data, performed the computation work, prepared figures and/or tables, and approved the final draft.
- Sabur Butt conceived and designed the experiments, performed the experiments, analyzed the data, prepared figures and/or tables, and approved the final draft.

- Hsien-Tsung Chang conceived and designed the experiments, performed the experiments, analyzed the data, authored or reviewed drafts of the paper, and approved the final draft.
- Grigori Sidorov conceived and designed the experiments, performed the experiments, authored or reviewed drafts of the paper, and approved the final draft.
- Alexander Gelbukh conceived and designed the experiments, performed the experiments, authored or reviewed drafts of the paper, and approved the final draft.

## Data Availability

Data, codes and other files such as the MEKA software feature files are available at GitHub: https://github.com/Noman712/Mutilabel_Emotion_Detection_Urdu.

## Supplemental Information

Supplemental information for this article can be found online at http://dx.doi.org/10.7717/peerj-cs.896#supplemental-information.

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
