# Peer review of "Multi-label emotion classification of Urdu tweets"

_PeerJ Computer Science, doi:10.7717/peerj-cs.896_

## Round 0.1 · original submission · Major Revisions

In general, the comments are quite positive, although some improvements are necessary for the paper. Please follow the reviewers' recommendations in the new version.

Reviewer 2 has requested that you cite specific references. You may add them if you believe they are especially relevant. However, I do not expect you to include these citations, and if you do not include them, this will not influence my decision.

Reviewer 1 ·

Basic reporting

please see my review below

Experimental design

please see my review below

Validity of the findings

please see my review below

Additional comments

The manuscript is centered on a very interesting and timely topic, which is also quite relevant to the themes of PeerJ Computer Science. Organization of the paper is good and the proposed method is quite novel. The length of the manuscript is about right but keyword list is missing. The paper, moreover, does not link well with recent literature on sentiment analysis appeared in relevant top-tier journals, e.g., the IEEE Intelligent Systems department on "Affective Computing and Sentiment Analysis". Also, latest trends in multilingual sentiment analysis are missing, e.g., see Lo et al.’s recent survey on multilingual sentiment analysis (from formal to informal and scarce resource languages). Finally, check recent resources for multilingual sentiment analysis, e.g., BabelSenticNet.

Authors seem to handle sentiment analysis simply as a binary classification problem (positive versus negative). What about the issue of neutrality or ambivalence? Check relevant literature on detecting and filtering neutrality in sentiment analysis and recent works on sentiment sensing with ambivalence handling.

Finally, the manuscript only cites a few papers from 2020 and 2021: check latest works on attention-based deep models for sentiment analysis and recent efforts on predicting sentiment intensity using stacked ensemble.

Some parts of the manuscript may result unclear for some readers of this journal. A short excursus on emotion categorization models and algorithms could resolve this lack of clarity (as the journal does not really feature many papers on this topic) and improve the overall readability of the paper. On a related note, the manuscript presents some bad English constructions, grammar mistakes, and misuse of articles: a professional language editing service is strongly recommended (e.g., the ones offered by IEEE, Elsevier, and Springer) to sufficiently improve the paper's presentation quality for meeting the high standards of PeerJ Computer Science.

Finally, double-check both definition and usage of acronyms: every acronym should be defined only once (at the first occurrence) and always used afterwards (except for abstract and section titles). Also, it is not recommendable to generate acronyms for multiword expressions that are shorter than 3 words (unless they are universally recognized, e.g., AI).

·

Basic reporting

Well written, but needs revisions.

Experimental design

NA

Validity of the findings

NA

Additional comments

Abstract and Introduction should be revised.
Problem statement must be clearly defined in the Introduction.
Use simple present tense throughout the paper.
Related work should have one paragraph of motivation due to limitations of existing approaches. Also, it should have references to the recent similar works.
Authors can consider referring the following articles:
A Consolidated Decision Tree-based Intrusion Detection System for binary and multiclass imbalanced datasets
Performance Assessment of Supervised Classifiers for Designing Intrusion Detection Systems: A Comprehensive Review and Recommendations for Future Research
Applications in Security and Evasions in Machine Learning: A Survey
Comparison of the work with an recent existing approach is necessary to show the performance improvement.
Result analysis must be thorough.
Conclusion should include limitations of the existing work.

Reviewer 3 ·

Basic reporting

- The use of the verb "talks" refers to Section 2 in line 64 of page 2 is not suitable.

- Spelling errors and punctuation errors are found in the second paragraph of page 9 (line 286-295).

- Section 4.3: I suggest use variables represent "number of c(e)" or "number of p(e)" in equations on page 10. The variable c(e) should be written in mathematical form (italic). This change should apply to all variables in this article.

- In Section 4.2, page 9: The use of capital letters in "We used the following BERT
301 parameters: MAXSEQLENGTH= 64, BATCH SIZE = 32, LEARNING RATE =2e5,
and NUM TRAIN EPOCHS= 2.0.

Experimental design

- The paper should explain the importance of all evaluation metrics in the context of this task. Explain how the metrics can describe the performance of the algorithms.

- Coding for the project is publicly available.

- In paragraphs 3 and 4 of section 5: The deep learning algorithms performed poorly in this study, why do authors claim that the algorithms perform well and the results are promising?

- In section 4.2, why the values of parameters were selected in this project.

Validity of the findings

- The description of the challenges faced in collecting information for this dataset and the annotation process should not be considered a contribution.

- No prior work on multilabel emotion classification exists for the Urdu language, and the generation of the dataset can be considered a contribution of this research.

- However, the performance of deep learning and machine learning algorithms for this task are very poor with low accuracy. Would you please explain why this is the case?

- I would suggest more evaluation be conducted to identify algorithms with better accuracy. Please justify the relevance of your findings even though the accuracy results are mostly low for all algorithms in this study.

---

## Round 0.2 · Major Revisions

Reviewer 1 has serious concerns about the acceptance of the paper and, after reading his comments, so do I. Please prepare a new version addressing all the suggestions given previously.

Reviewer 1 ·

Basic reporting

Many of the claims made by the authors are not backed by the revision, e.g., author claim to have added literature related to neutrality and ambivalence for sentiment analysis but there is no mention whatsoever of neither of them. Also, authors claim to have fixed acronyms but that is simply not true. Finally, presentation is still not up to PeerJ standards and important relevant literature on multilingual sentiment analysis is still missing.

Experimental design

Please refer to my previous review

Validity of the findings

Please refer to my previous review

---

## Round 0.3 · accepted · Accept

According to the reviewer's opinion, the article is now acceptable for publication. I agree with them

Reviewer 1 ·

Basic reporting

The authors have addressed all of my concerns and their revisions have substantially improved the manuscript.

Experimental design

Good.

Validity of the findings

Good.